# The Genes—Candidates for Prognostic Markers of Metastasis by Expression Level in Clear Cell Renal Cell Cancer

**DOI:** 10.3390/diagnostics10010030

**Published:** 2020-01-08

**Authors:** Natalya Apanovich, Maria Peters, Pavel Apanovich, Danzan Mansorunov, Anna Markova, Vsevolod Matveev, Alexander Karpukhin

**Affiliations:** 1Bochkov Research Centre for Medical Genetics, 1 Moskvorechye St., Moscow 115522, Russia; apanovichn81@mail.ru (N.A.); apanovich2004@mail.ru (P.A.); gah3ah@mail.ru (D.M.); 2N.N. Blokhin National Medical Research Center of Oncology of the Ministry of Health of Russia, 24 Kashirskoe Shosse, Moscow 115478, Russia; omega0803@mail.ru (M.P.); mark-an1@yandex.ru (A.M.); vsevolodmatveev@mail.ru (V.M.)

**Keywords:** renal cancer, gene expression, markers

## Abstract

The molecular prognostic markers of metastasis are important for personalized approaches to clear cell renal cell carcinoma (ccRCC) treatment but markers for practical use are still missing. To address this gap we studied the expression of ten genes—*CA9*, *NDUFA4L2*, *VWF*, *IGFBP3*, *BHLHE41*, *EGLN3*, *SAA1*, *CSF1R*, *C1QA*, and FN1—through RT-PCR, in 56 ccRCC patients without metastases and with metastases. All of these, excluding *CSF1R*, showed differential and increased (besides *SAA1*) expression in non-metastasis tumors. The gene expression levels in metastasis tumors were decreased, besides *CSF1R*, *FN1* (not changed), and *SAA1* (increased). There were significant associations of the differentially expressed genes with ccRCC metastasis by ROC analysis and the Fisher exact test. The association of the *NDUFA4L2*, *VWF*, *EGLN3*, *SAA1*, and *C1QA* expression with ccRCC metastasis is shown for the first time. The *CA9*, *NDUFA4L2*, *BHLHE4*, and *EGLN3* were distinguished as the strongest candidates for ccRCC metastasis biomarkers. We used an approach that presupposed that the metastasis marker was the expression levels of any three genes from the selected panel and received sensitivity (88%) and specificity (73%) levels with a relative risk of RR > 3. In conclusion, a panel of selected genes—the candidates in biomarkers of ccRCC metastasis—was created for the first time. The results might shed some light on the ccRCC metastasis processes.

## 1. Introduction

Almost half of the patients with clear cell renal cell carcinoma (ccRCC) develop metastases [1]. Even after surgical treatment of early non-metastatic kidney cancer, metastases occur in 5–15% of cases within five years of follow-up, which significantly affect the survival of patients [2]. The frequency of metastasis can be about three-fourths of all relapse cases and is weakly associated with some clinical characteristics [3]. These factors make it necessary to predict the occurrence of metastases. However, indicators that predict the risk of metastasis are not available [4]. Information about the risk of metastasis could be taken into account with the possible appointment of postoperative therapy. In addition, a biopsy of small tumor formations in the kidney is being actively developed [5]. The identification of tumors prone to metastasis can be taken into account when determining the volume of surgical intervention, including the removal of small tumors. Therefore, the need to predict their occurrence is very relevant. The presence of prognostic markers will create the ability to form groups for dynamic observation, apply therapeutic approaches to prevent metastasis, detect emerging metastases in time, and treat patients effectively [6]. At the same time, molecular markers of the tumor metastatic potential used in practice are still missing [7].

The mechanisms of metastasis are still not well-understood [8]. Currently, some genes associated with the ccRCC metastasis are known [4,9,10,11], but the possibility of their use as markers is not shown [4]. The identification of such markers could enable personalized approaches to postoperative treatment, as well as contribute to an understanding of the molecular pathways for targeted therapeutic interventions [8,12].

In order to search for connections within tumor metastasis genes, we studied the expression profile of ten genes that might be significant for the development of kidney cancer—*CA9*, *NDUFA4L2*, *VWF*, *IGFBP3*, *BHLHE41*, *EGLN3*, *SAA1*, *CSF1R*, *C1QA*, and *FN1*, in tissue samples of clear cell carcinoma of the kidney. For analysis, two groups were identified—with the absence of metastases and with synchronous metastases.

## 2. Materials and Methods

The samples were obtained at the “N.N. Blokhin Medical Research Center of Oncology” (N. N. Blokhin MRSO). All subjects gave their informed consent for inclusion before they participated in the study. The study was conducted in accordance with the Declaration of Helsinki, and the protocol was approved by the Ethics Committee of the Federal State Budgetary Institution “Research Centre for Medical Genetics” (the approval number 2017-4/2). All samples underwent histological examination in the Department of Pathological Anatomy of Human Tumors at N. N. Blokhin MRSO, and were clinically characterized. A total of 56 paired samples of kidney tissue (tumor tissue and morphologically normal tissue of the same kidney) were examined. Among them, there was a group with the absence of ccRCC metastases during the observation period of more than 3 years (22 patients). The group with metastases included patients who only had synchronous metastases, i.e., the metastases were identified in them within 1 year after the main diagnosis—clear cell renal cell carcinoma (34 patients).

The choice of genes for the analysis of the relationship of their expression with metastasis was carried out in two stages. Initially, using the databases GEO (available online: www.ncbi.nlm.nih.gov/geo, accessed on 6 January 2020) and Oncomine (available online: www.ocomine.org, accessed on 6 January 2020), more than 15 studies of gene expression were analyzed, each of which included at least 20 samples. In each study, genes with the highest expression relative to normal tissue were isolated. When selecting the genes, the criterion of increased gene expression in more than 50% of the analyzed works was used. In this way, 200 genes were selected.

At the second stage, a screening analysis of the expression of these genes was performed on a sample of 20 paired samples of tumor tissue and morphologically normal tissue using the Real-Time-PCR method. As a result, 20 genes were selected for subsequent analysis (Figure 1). The criterion for the selection of genes was expression, increased by more than 2 times, in more than 50% of the ccRCC samples.

The mRNA expression levels of genes in the tumor tissue were determined from histologically normal paired kidney tissues that were distant from the tumor. For isolation of total RNA from the surgical material of tumor and normal kidney tissue, the RNeasy Mini Kit (QIAGEN, Germantown, MD, USA) was used. The quality of obtained total RNA was checked using electrophoresis in 1.8% agarose gel (in a chamber for horizontal electrophoresis SUB CELL 96 BIO-RAD (Temecula, CA, USA)). The intensity of the bands (28S/18S) after EtBr staining was estimated using the gel documentation system Gel Doc XR + BIO-RAD (USA). Measurement of the RNA concentration, as well as an assessment of the purity of RNA by the ratio between the readings taken at 260 nm and 280 nm (260/280), was carried out on a Nanodrop 1000 spectrophotometer (Thermo Fisher Scientific, Wilmington, DE, USA). The RNA was considered to be qualitative in the presence of two bands that were clearly visible, corresponding to the 28S and 18S (in this case, the 28S band should be approximately twice as intense as the 18S band), as well as with the ratio A260/A280, equal to 1.8–2.1. The reverse transcription reaction was performed using the M-MLV Reverse Transcriptase kit (Life Technologies, Carlsbad, CA, USA). The quantitative real-time polymerase chain reaction (RT-PCR) with reverse transcription products was performed using kits from Applied Biosystems, USA: TaqMan^®^ Gene Expression Master Mix (2-fold universal reaction mixture) and TaqMan^®^ Gene Expression Assay (reaction mixture of primers and fluorescently-labeled oligonucleotide) to determine each gene. The RT-PCR for each gene and sample was carried out in triplicates. A negative control contained a PCR mixture without a matrix. As an endogenous control, the *GAPDH* gene was used [13]. Relative expression was calculated using Step One Software as the ∆∆Ct (RQ) method.

Statistical data processing was performed using the software Statistica 10.0, MedCalc program and the online calculator: https://www.medcalc.org/calc/diagnostic_test.php (accessed on 6 January 2020). Differences in the expression levels were evaluated using the U criterion; ROC analysis, Fisher’s exact test, and the logistic regression method were used to evaluate the relationship between the expression levels and metastasis.

Since we conducted a study on the association between metastasis and simultaneous expression of several genes, we applied the correction for the multiplicity of comparisons using the false discovery rate method (FDR) [14]. The application of this amendment avoids false “discoveries” that might arise for statistical reasons in multiple comparisons. The significance level was taken to be equal to less than 0.05.

## 3. Results

Figure 1 shows the expression of 20 genes, selected as the most highly and frequently expressed, according to the results of a screening analysis of the expression of 200 genes in the ccRCC samples.

For all studied genes, a tendency toward expression changes with the progression of ccRCC was visible. This might indicate the possible significance of these genes for metastasis. For the initial study, 9 genes with different expression levels were selected. The *SAA1* gene was added to this list, as the literature suggests an increase in its expression in the third stage of ccRCC [15]. Thus, a sample of 10 genes was formed to study the relationship of their expression with metastasis of ccRCC—*CA9*, *NDUFA4L2*, *VWF*, *IGFBP3*, *BHLHE41*, *EGLN3*, *SAA1*, *CSF1R*, *C1QA*, and *FN1*.

Among the studied genes, expression levels higher than in normal tissues with non-metastatic tumors was found for most genes, except *SAA1* and *CSF1R*. The expression of the *FN1* was slightly increased. In metastatic tumors, most genes had lower expression levels relative to tumors without metastases. The *CSF1R* and *FN1* genes did not show statistically significant changes in expression, and the *SAA1* gene showed higher expression under tumor metastasis (Table 1, Figure 2).

To characterize the relationship between the expression levels of these genes with metastasis, ROC analysis was used.

The analysis revealed that the expression of 8 out of 10 analyzed genes had a statistically significant relationship with metastasis of the ccRCC (Table 2). The significance of the differences was retained when applying the Benjamini–Hochberg procedure for multiple comparisons (FDR). That is, a decrease in the expression level of the *CA9*, *NDUFA4L2*, *VWF*, *IGFBP3*, *BHLHE41*, *EGLN3*, and *C1QA* genes was an unfavorable prognosis for the development of metastases. For the *SAA1* gene, an unfavorable prognosis was associated with an increase in its expression level.

Consequently, differences in the values of median expression levels and ROC analysis indicated a relationship between the expression of the eight studied genes and the metastasis of the ccRCC. Further, to quantify this relationship, the odds ratio (OR) and relative risk (RR) values were obtained, and the association of gene expression levels with metastasis was determined using the Fisher exact test. In the calculations, we used the cut-off value of the expression levels of the studied genes (Table 2) determined by the ROC analysis, which optimally shared the distribution of the expression levels for groups with and without metastases and the best sensitivity and specificity levels (Table 2). The expression frequency above/below the cut-off value was determined in two groups with and without synchronous metastases (Table 3).

From the data in Table 3, it can be seen that the same genes that were detected by the ROC analysis turned out to be significant for the development of metastases, in accordance with the exact Fisher criterion and 95% CI for OR and RR. For genes with a statistically significant relationship with metastasis, the OR for different genes was in the range of about 6 to 70, and the RR was in the range of 1.5 to 8, with a minimum value of 95% CI greater than 1. The highest values of OR (more than 10) and RR (from approximately 2.5) were found for the *CA9*, *NDUFA4L2*, *BHLHE41*, *EGLN3*, and *VWF* genes.

Additionally, the logistic regression method was used. In accordance with the results of this method, the genes that were significantly associated with metastasis were—*CA9*, *NDUFA4L2*, *BHLHE41*, *EGLN3*, *VWF*, and *IGFBP3*. That is, the expression levels of these genes in metastatic and non-metastatic ccRCC tumors showed the property of linear separability. The *C1QA* and *SAA1* gene expression levels associated with metastasis, according to the results of other statistical methods, did not have this property (Table 3).

According to the set of traits—AUC > 0.8 (Table 2), the highest OR values and linear separability of the expression levels between metastatic and non-metastatic tumors (Table 3), four genes could be distinguished as the strongest candidates for ccRCC metastasis biomarkers—*CA9*, *NDUFA4L2*, *BHLHE41*, and *EGLN3*. For these genes, the sensitivity indicators were in the range of 79–97% and the specificity was in the range of 68–91% (Table 2). The expression level of these genes above the cut-off value (Table 2) was a favorable factor, and an expression level below this value was an unfavorable factor.

The following approaches were used to evaluate the practical applicability of these results: early metastasis marker are expression levels of (a) two genes and (b) three genes from the panel below the cut-off value. Following approach (a), a maximum sensitivity of 100% and a high OR value were obtained, indicating a close relationship between such a marker and metastasis. Approach (b) led to a slight decrease in sensitivity but an increase in specificity was observed. Relative risk (RR) in both cases was near region 3 (Table 4).

## 4. Discussion

Thus, the expression levels of eight genes were associated with the metastasis of ccRCC—*CA9*, *NDUFA4L2*, *VWF*, *IGFBP3*, *BHLHE41*, *EGLN3*, *SAA1*, and *C1QA*. Five of these genes—*CA9*, *NDUFA4L2*, *IGFBP3*, *BHLHE41*, and *EGLN3* were direct targets of HIF1α [16,17,18,19,20]. The HIF1α acted as a transcriptional regulator that organizes cell adaptation to the microenvironment with low oxygen levels, for example, due to the growth of solid tumors [21]. The accumulation of HIF1α also occurs due to inactivation of the *VHL* gene. Adaptation is accompanied by the expression of genes and is characterized by the transition of homeostasis regulation to a new level. The induction of the expression of genes activated by hypoxia is caused by the interaction of the transcription factor with a characteristic DNA region (an element of the response to hypoxia, HRE) located in the regulatory regions of such genes. Transcription factors control the binding of RNA polymerase II to the gene promoter and thereby provide control of mRNA synthesis [22]. The increase in the expression level of these five genes is apparently due to the accumulation of HIF1α and is associated with the processes of tumor cell adaptation to hypoxia. It can be assumed that the processes associated with a strong initial response to stress and hypoxia goes out as the ccRCC tumor progresses. At the same time, other processes develop.

In particular, for the first time we have shown the importance of the *C1QA* gene in the development of kidney cancer. This gene is differentially expressed in ccRCC tumors and its expression level is associated with tumor metastasis. The C1QA is a subunit of the C1 complement complex, which, in turn, consists of three proteins—C1q, C1r, and C1s. The interaction of C1q with immune complexes leads to complement activation [23]. C1q expression is induced by hypoxia [24]. In addition to the role of the complement initiator, C1q is also involved in other processes that are completely independent of the complement activation [25]. In particular, C1q can induce destabilization of cell adhesion [26]. It promotes cell adhesion by interacting with fibronectin, one of the key proteins of the intercellular matrix [27]. This C1q function can explain the effect of its expression on metastasis—a decrease in the level of C1q expression can lead to a weakening of intercellular interactions and, consequently, to the separation of tumor cells and their migration.

As evidenced by the observed increase of *SAA1* expression in the metastasizing tumors, another developing process might be inflammatory. The product of the *SAA1* gene is an acute inflammation phase protein [28]. An increase of this gene expression level might be associated with a chronic inflammatory process that occurs during the progression of kidney cancer [29]. Increased expression of *SAA1* was found in an aggressive form of ccRCC [30]. An increase in serum SAA1 was detected at stage III of ccRCC [15].

Under conditions of hypoxia, HIF-1α induces transcription of *BHLHE41* (*DEC2*), coding basic helix–loop–helix, bHLH, a transcription factor whose expression is regulated by hypoxia [31]. Some of the functions of this gene are cell differentiation, the immune response, regulation of the molecular clock, and carcinogenesis [32]. A significant increase in the expression of *BHLHE41* in kidney cancer was observed in studies under the TCGA [33]. Increased expression of *BHLHE41* stimulated the proliferation of tumor cells of ccRCC, and a knockdown led to a significant decrease in this feature [34]. A negative correlation between the expression level of *BHLHE41* and tumor invasion, lymph node metastases, TNM stage, and poor survival, was found among patients with gastric cancer. Increased expression of *BHLHE41* led to the inhibition of EMT in vitro, as well as tumor growth and in vivo metastasis. The anti-metastatic effect of *BHLHE41* was mediated by the inhibition of the ERK/NF-κB /EMT pathway [32]. In breast cancer, an increase in *BHLHE41* also inhibited metastasis and proliferation [35,36,37]. In addition, *BHLHE41* disrupts the epithelial–mesenchymal transition (EMT) in human endometrial and pancreatic cancer [38,39]. The well-known mechanisms of the *BHLHE41* action correspond to the relationship found between the expression level of this gene and the metastasis of ccRCC. Decreased expression of *BHLHE41* might contribute to the development of metastases by activating the epithelial–mesenchymal transition (EMF). A decrease in its expression is an unfavorable prognostic factor.

The *NDUFA4L2* gene has a pronounced expression dependence on hypoxia. It is a direct target of the *HIF1α* gene [40]. The importance of NDUFA4L2 expression for the development of ccRCC was first shown by us [41] and then in other papers [42,43,44]. In addition to ccRCC, this gene is overexpressed in non-small cell lung cancer (NSCLC), as well as in NSCLC cell lines cultured under hypoxia. Wherein, activated by HIF1α, *NDUFA4L2* inhibited ROS production by the respiratory chain of mitochondria in non-small cell lung cancer cells. Knockdown of *NDUFA4L2* promoted an increase in ROS production, apoptosis, and the progression of EMT in the NSCLC cell lines [45]. Apparently, the activation of EMT due to a decrease in the expression level of *NDUFA4L2* is a mechanism of its influence on metastasis of ccRCC.

The prolyl hydroxylase 3 gene, *EGLN3* (*PHD3*), encodes a hydroxylase whose expression is HIF1α dependent [46]. Screening studies and our data have shown a differential expression of this gene in ccRCC [47,48]. According to our results, a decreased level of the *EGLN3* gene expression is associated with an increased risk of metastasis.

Another direct target of HIF-1α is the *IGFBP3* gene [16]. Its product, IGFBP3, is a multifunctional protein that modulates the activity of the IGF/IGF–IR system and plays an important role, both in the activation of cell proliferation and in its inhibition [49]. It is known that *IGFBP3* can be considered as one of the potential markers of ccRCC, a significant increase of its expression is found in clear cell renal cancer [50,51]. It has been suggested that elevated serum *IGFBP3* concentrations might protect against cancer [52,53]. Increased expression of *IGFBP3* inhibits the growth of metastatic cells in the lungs, but does not inhibit the growth of primary tumor cells and normal kidney cells [54]. *IGFBP3* has a different effect on cell proliferation—it stimulates the growth of a primary tumor while inhibiting metastases [55]. Our data indicated an increased expression of *IGFBP3* in tumors relative to the norm in non-metastatic ccRCC. Its decrease was associated with metastasis.

With respect to the genes already considered above, the expression of the *CA9* gene was studied most thoroughly in ccRCC. The product of this gene, type 9 carboxyanhydrase, is a transmembrane and cytosolic glycoprotein, which is involved in the regulation of pH and acid–base balance in the body [17,56]. Bui et al. showed a decrease of the CA9 IHC staining of metastases in relation to the corresponding primary tumor samples. The results of the study also showed that a decrease in CA9 expression occurs in tumors with the highest potential for malignancy. This suggests that CA9 might play a functional role in tumor progression [57]. The meta-analysis showed a relationship between CA9 expression and ccRCC clinical characteristics. Zhao Z. et al. note that the low level of CA9 expression correlates with a number of ccRCC clinical features, including, a high degree of differentiation, the presence of metastases in lymph nodes, and distant metastases [58].

The data cited above corresponded to our results—the level of *CA9* gene expression in our work also decreased in the presence of metastases. That is, a decrease in its expression indicated an unfavorable prognosis.

Von Willebrand Factor (VWF) is a plasma protein. It mediates platelet adhesion to the damaged vessel wall, and also transfers and protects coagulation factor VIII. At the site of vascular damage, VWF binds to unprotected collagen, thereby promoting platelet adhesion [59]. Increased expression of the *VWF* gene with respect to normal tissue was noted in a meta-analysis of kidney cancer expression data [47]. Higher expression of the *VWF* gene was associated with a better survival of patients with ccRCC [60]. Our results revealed a correlation between low levels of *VWF* expression and metastasis of ccRCC.

Therefore, the functional features of a number of genes associated with metastasis of ccRCC revealed in this work indicated the complex nature of the processes that occur during metastasis. These can be expressed by inducing an EMT (*BHLHE41*, *NDUFA4L2*), a decrease in the adhesive properties of tumor cells (*C1QA*), and activation of inflammatory reactions (*SAA1*). The mechanisms of influence on the metastasis of other identified genes are not yet clear. However, a decrease in expression upon metastasis of all HIF-1α-dependent genes studied in this work is of interest. This might imply a completion of the adaptation process and a transition of the tumor cells to the next, more malignant condition.

There are some publications on the associations between gene expression and metastasis. Yang et al. conducted an analysis of gene expression on microarrays of four tissue samples of kidney cancer without metastasis and five with early metastasis [10]. The received data was not validated. Expression profiles were also compared on kidney cancer samples and lung metastases using Affymetrix microarrays to identify genes that could potentially be targets of therapy [61]. Gene expression profiles were studied using microarrays on cultured metastatic RCC (ACHN) and primary RCC (CAKI-2) cell lines, to identify and analyze cancer stem cells [62]. In the works presented here, as well as in the works devoted to the analysis of the expression of individual genes [4,9,10,11], missing information on the applicability of the revealed differences in expression as markers of metastasis.

From the point of view of using expression levels as metastasis biomarkers, the most promising genes are *CA9*, *NDUFA4L2*, *BHLHE41*, and *EGLN3*. According to the set of characteristics studied, they are the best classifiers. As an approximation to the practical applicability of these biomarkers, we examined the prognostic characteristics obtained by simultaneously taking into account the expression levels of three of these four genes. This approach minimized the effect of experimental errors in determining expression levels. Acceptable values of sensitivity (88%) and specificity (73%) for detecting metastatic ccRCC tumors with a relative risk of RR > 3 were obtained. These results opened the way to creating a prognostic formula for metastasis of ccRCC in further studies on extended samples. Preliminary analysis showed that in 70% of cases, the use of the created panel allowed for obtaining a definite result on the tumors’ tendency to metastasize. It was obtained by presupposing the following—favorable prognosis (metastases will not develop within 1 year)—if the expression level was above the threshold in at least three genes; poor prognosis (metastases will develop within 1 year)—if the expression level was below the threshold in four genes. Gray area—30%. There were no false results.

In conclusion, among the investigated genes, the significance of the *NDUFA4L2*, *VWF*, *EGLN3*, *SAA1*, and *C1QA* in the expression for metastasis of ccRCC was for the first time shown in our work. The four promising genes—candidates for prognostic markers of metastasis—were identified. The level of these gene expressions in the group without metastases was increased. A decrease in their expression level in the group with synchronous metastases was associated with a poor prognosis of metastases development. These four genes can form a panel that allows predicting the risk of metastasis with fairly high sensitivity and specificity values. Such a panel is formed for the first time for ccRCC metastasis. So as these genes—*CA9*, *NDUFA4L2*, *BHLHE41*, and *EGLN3*—are HIF-1α-regulated, from the standpoint of the metastasis biology, a decrease in their expression might suggest a transition of tumor cells to a more malignant condition after an adaptation process that might provoke metastases. In addition, *BHLHE41* and *NDUFA4L2* might induce EMT.

## Figures and Tables

**Figure 1 diagnostics-10-00030-f001:**
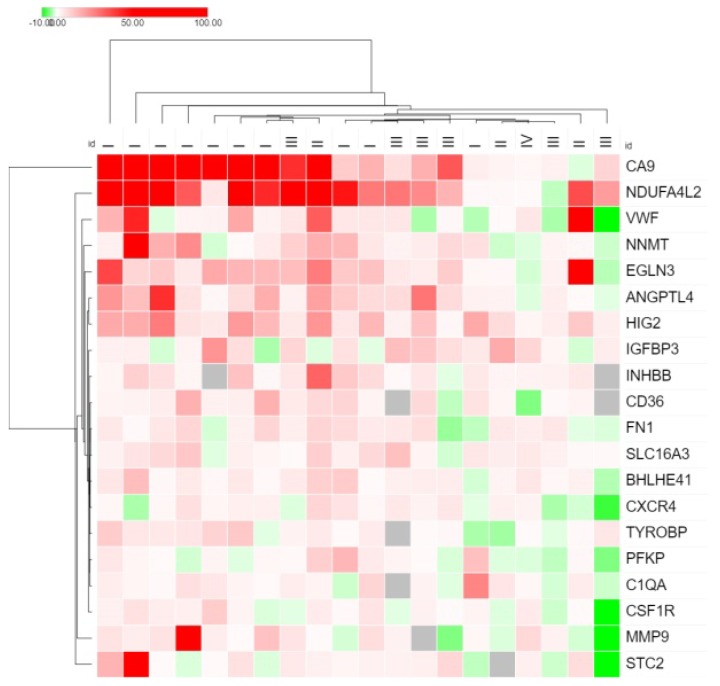
Heat map of the expression of 20 genes in the clear cell renal cell carcinoma (ccRCC) samples. At the top of the figure, the stages of cancer development for each sample are shown.

**Figure 2 diagnostics-10-00030-f002:**
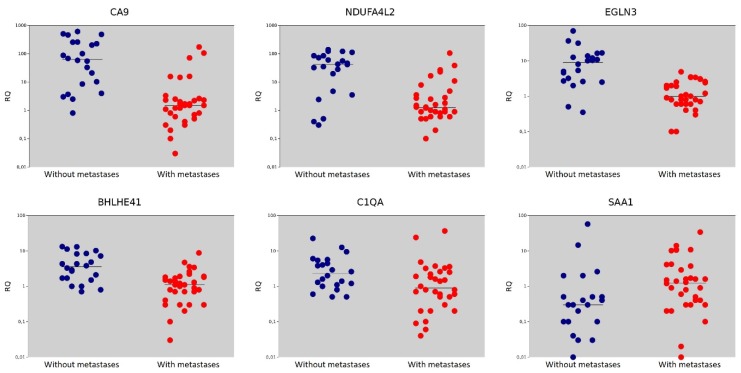
Relative gene expression (RQ) in groups with metastases (●) and without metastases (●). Gene expression values are presented on a logarithmic scale. The line marks the median.

**Table 1 diagnostics-10-00030-t001:** The values of the medians and the significance of their differences in the ccRCC groups of with and without metastases.

Gene	Gene Name	The Median Value in the Non-Metastasis Group	The Median Value in the Metastasis Group	*p* = (Mann–Whitney U-test)
*CA9*	Carbonic Anhydrase 9	63.15	1.50	<0.001
*NDUFA4L2*	NADH Dehydrogenase [Ubiquinone] 1 Alpha Subcomplex Subunit 4-Like 2	42.00	1.25	<0.001
*EGLN3*	Egl Nine Homolog 3	9.00	1.00	<0.001
*BHLHE41*	Basic Helix-Loop-Helix Family Member E41	3.55	1.10	<0.001
*C1QA*	Complement C1q Subcomponent Subunit A	2.30	0.90	0.016
*SAA1*	Serum Amyloid A1	0.30	1.25	0.018
*CSF1R*	Colony Stimulating Factor 1 Receptor	0.95	0.90	0.562
*FN1*	Fibronectin	1.30	1.15	0.396
*VWF*	Von Willebrand Factor	2.60	0.75	<0.001
*IGFBP3*	Insulin Like Growth Factor Binding Protein	2.55	1.10	0.007

**Table 2 diagnostics-10-00030-t002:** The relationship of gene expression with the development of metastases—ROC analysis.

Gene	Area under ROC Curve (AUC)	95% CI	Cutoff Value	Significance Level *p* (Area = 0.5)	Sensitivity	Specificity
*CA9*	0.891	0.804–0.978	≤2.6	<0.001	79.41	90.91
*NDUFA4L2*	0.811	0.672–0.951	≤27	<0.001	94.12	68.18
*VWF*	0.788	0.659–0.917	≤1.4	<0.001	70.59	81.82
*BHLHE41*	0.808	0.691–0.925	≤2.6	<0.001	88.24	63.64
*EGLN3*	0.873	0.760–0.986	≤3.5	<0.001	97.06	68.18
*IGFBP3*	0.714	0.572–0.856	≤1.7	0.003	73.53	68.18
*SAA1*	0.689	0.537–0.841	>0.5	0.015	64.71	77.27
*CSF1R*	0.547	0.389–0.705	≤2.1	0.562	-	-
*C1QA*	0.693	0.552–0.833	≤3.7	0.007	91.18	40.91
*FN1*	0.568	0.412–0.724	≤0.8	0.198	-	-

**Table 3 diagnostics-10-00030-t003:** The gene expression level frequencies relative to the cut-off value in groups of patients with metastases and without ccRCC metastases and the characteristics of the association with metastasis.

Gene	Frequency of Expression Higher/Lower from the Cut-Off Value in the Non-Metastasis Group	Frequency of Expression Higher/Lower from the Cut-Off Value in the Metastasis Group	Odds Ratio/95% CI	Relative Risk/95%CI	Fisher Exact Two-Tailed, *p* =	Logistic Regression, *p* =
*CA9*	20/2	7/27	38.57/7.23–205.82	8.74/2.30–33.11	<0.001 *	<0.001 *
*NDUFA4L2*	15/7	2/32	34.29/6.35–185.24	2.96/1.60–5.48	<0.001 *	<0.001 *
*VWF*	18/4	10/24	10.80/2.91–40.06	3.88 1.56–9.67	<0.001 *	<0.001 *
*BHLHE41*	14/8	4/30	13.13/3.38–51.01	2.43/1.38–4.27	<0.001 *	<0.001 *
*EGLN3*	15/7	1/33	70.71/7.97–627.07	3.05/1.65–5.64	<0.001 *	<0.001 *
*IGFBP3*	15/7	9/25	5.95/1.83–19.31	2.31/1.21–4.40	0.003 *	0.005 *
*SAA1*	5/17	22/12	6.23/ 1.84–21.12	2.85/1.27–6.40	0.003 *	0.967
*CSF1R*	7/15 (31.8)	5/29 (14.7)	2.71/0.73–10.00	1.25/0.91–1.72	0.184	0.421
*C1QA*	9/13 (40.9)	3/31 (8.8)	7.16/1.66–30.75	1.54/1.07–2.21	0.007 *	0.540
*FN1*	16/6 (72.7)	18/16 (52.9)	2.37/0.75–7.52	1.73/0.80–3.73	0.169	0.153

* significant with false discovery rate (FDR) correction for multiple comparisons.

**Table 4 diagnostics-10-00030-t004:** Relationship of simultaneous expression of several genes with the development of ccRCC metastasis within 1 year.

Gene Group	Sensitivity/Specificity	Odds Ratio/95% CI	Relative Risk/95%CI	Negative Predictive Value %/95% CI
Two genes of *CA9* *BHLHE41* *EGLN3* *NDUFA4L2*	100.00%/68.18%	142.60/7.66–2656.78	3.14/1.70–5.79	100.00
Three genes of *CA9* *BHLHE41* *EGLN3* *NDUFA4L2*	88.24%/72.73%	20.00/4.92–81.36	3.24/1.62–6.47	80.00/60.61–91.23

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
