# Peer review of "The Genes—Candidates for Prognostic Markers of Metastasis by Expression Level in Clear Cell Renal Cell Cancer"

_diagnostics, 2020, doi:10.3390/diagnostics10010030_

Round 1
Reviewer 1 Report
The authors should include few sentences and public health data showing the importance of the identification of prognostic markers of metastasis in ccRCC.
The topic is very important but it was not explained the rationale and relevance of this paper.
Also, few papers have been published in the topic; so they should put more emphasis of the novelty of this particular job. In addition, they should comment in the discussion their results with the previously reported.
Please add a column in the tables to describe the name of each gene; that will make more sense to the readers when they read the results.
In statistical material and methods, the authors should described the FDR correction for multiple comparisons and why they have done that.
Please correct the whole paragraph in Russian to English (page 6-7).
The author described the importance of each gene in the discussion; but they did not elaborate an explanation of why those 5 particular significant genes re important for the stand point of the metastasis biology.
Author Response
Dear Reviewer,
Thank you very much for useful recommendations to improve the article.
All of these are accounted for:
importance of the identification of prognostic markers of metastasis in ccRCC is explained reinforced indications of the work novelty; the results are commented in the discussion with the previously reported data the name of each gene is added in the table 1; unfortunately, it is impossible to insert into other tables for technical reasons (low space) the FDR correction is described in methods the paragraph in Russian is deleted an explanation of why particular significant genes are important for the stand point
of the metastasis biology is added in the conclusion
Gratefully,
Alexander Karpukhin
Reviewer 2 Report
With current RNAseq or even single cell RNAseq technology present, traditional QPCR method to evaluate gene expression profile is limited. The author should further explain the reason to choose these ten genes in the method part. And this bioinformatics analysis should be an important part of result as well. Images from histology analysis is also needed to distinguish tumors with or without metastases.
Author Response
Dear Reviewer,
Thank you very much for useful recommendations to improve the article.
All of these are accounted for:
the reason to choose the genes in the method and in result parts is explained the text in all sections of the article is improved unfortunately, it is impossible to distinguish tumors with or without metastases by histology images; we could not find the relevant information in the available literature of course, NGS and other modern methods of massive analysis provide essential information; at the same time, in the case of consideration of specific genes, especially for diagnostic or prognostic purposes, the data obtained by such methods need to be validated both when determining mRNA or mutations. RT-PCR is most suitable for analyzing gene expression in this case.
Gratefully,
Alexander Karpukhin
Round 2
Reviewer 2 Report
the authors gave explanations and they are acceptable.